# Cellular eEF1G Inhibits Porcine Deltacoronavirus Replication by Binding Nsp12 and Disrupting Its Interaction with Viral Genomic RNA

**DOI:** 10.3390/v17101369

**Published:** 2025-10-13

**Authors:** Weijia Yin, Xinna Ge, Lei Zhou, Xin Guo, Jun Han, Yongning Zhang, Hanchun Yang

**Affiliations:** 1State Key Laboratory of Veterinary Public Health Safety, College of Veterinary Medicine, China Agricultural University, Beijing 100193, China; 2Key Laboratory of Animal Epidemiology of Ministry of Agriculture and Rural Affairs, College of Veterinary Medicine, China Agricultural University, Beijing 100193, China

**Keywords:** porcine deltacoronavirus (PDCoV), eukaryotic elongation factor 1 gamma (eEF1G), nonstructural protein 12 (Nsp12), genomic RNA, replication

## Abstract

Porcine deltacoronavirus (PDCoV) is an emerging pathogen that causes severe, often fatal, diarrhea in suckling piglets and has zoonotic potential. Its nonstructural protein 12 (Nsp12), functioning as the RNA-dependent RNA polymerase (RdRp), is a central component of the viral replication–transcription complex and a critical target for host antiviral mechanisms. Here, we identified eukaryotic elongation factor 1 gamma (eEF1G) as a host interactor of PDCoV Nsp12 by immunoprecipitation-coupled mass spectrometry in IPEC-J2 cells. This interaction was confirmed by co-immunoprecipitation, pull-down assays, and confocal microscopy. Functional analyses involving siRNA knockdown and overexpression of eEF1G, combined with viral titration, strand-specific real-time quantitative PCR, and RNA immunoprecipitation assays, demonstrated that eEF1G directly binds to Nsp12. Knockdown of eEF1G significantly enhanced viral replication and increased negative-stranded RNA synthesis, whereas overexpression did not affect viral proliferation. Furthermore, eEF1G was found to bind PDCoV genomic RNA and competitively disrupt the interaction between Nsp12 and viral RNA, thereby impairing RdRp activity. Our results indicate that eEF1G acts as a novel host restriction factor that inhibits PDCoV replication by competing with Nsp12 for genomic RNA binding, ultimately blocking negative-stranded RNA synthesis. This study unveils a new antiviral mechanism and highlights a potential target for developing interventions against PDCoV.

## 1. Introduction

Coronaviruses (CoVs) constitute a significant threat to global public and animal health. Over the past two decades, the emergence of highly pathogenic zoonotic CoVs, including severe acute respiratory syndrome coronavirus (SARS-CoV), Middle East respiratory syndrome coronavirus (MERS-CoV), and severe acute respiratory syndrome coronavirus 2 (SARS-CoV-2), has precipitated epidemics with severe consequences for human morbidity and mortality [1,2]. Concurrently, CoVs pose a substantial risk to livestock health and economic stability, particularly within the swine industry. Currently, four distinct swine enteric CoVs are known to cause acute gastroenteritis and diarrheal disease in pigs: transmissible gastroenteritis virus (TGEV), porcine epidemic diarrhea virus (PEDV), porcine enteric alphacoronavirus (PEAV), and porcine deltacoronavirus (PDCoV) [3]. These viruses have led to substantial economic losses for the global pig industry. Among these, the emerging PDCoV is of particular concern. First identified in 2012 from porcine fecal samples in Hong Kong, China [4], PDCoV has since achieved a global distribution [5,6,7,8,9]. It primarily infects newborn piglets, causing severe diarrhea, vomiting, dehydration, and frequently leading to high mortality rates [10,11]. Despite demonstrating a high degree of genetic sequence conservation among global isolates, PDCoV exhibits a broad host tropism, being capable of infecting multiple animal species [12]. Most notably, and of significant public health relevance, is the report of a PDCoV spillover event into a human host, suggesting this virus possesses a potential and unexpected zoonotic capacity [13].

PDCoV is an enveloped virus with a single-stranded, positive-sense RNA genome, classified within the genus *Deltacoronavirus*, family *Coronaviridae*, order *Nidovirales* [14]. The approximately 25.5 kb genome is organized with two large, overlapping open reading frames (ORF1a and ORF1b) at its 5′ terminus. These ORFs are translated via a ribosomal frameshift mechanism into two polyprotein precursors, pp1a and pp1ab, which are subsequently proteolytically processed into 15 mature nonstructural proteins (Nsps) [14,15]. These Nsps, including key enzymes such as Nsp3 (papain-like protease), Nsp5 (3C-like protease), Nsp12 (RNA-dependent RNA polymerase, RdRp), are primarily responsible for viral replication and transcription. The 3′ terminus of the PDCoV genome encodes four essential structural proteins—spike (S), envelope (E), membrane (M), and nucleocapsid (N)—as well as three accessory proteins (NS6, NS7 and NS7a) [15]. The primary function of the Nsps is to assemble the replication and transcription complex (RTC), which directs the synthesis of viral genomic RNA and the transcription of subgenomic mRNAs [16]. Central to this complex is Nsp12, the catalytic RdRp subunit, which recognizes nucleotide triphosphates and catalyzes RNA synthesis in a 5′ to 3′ direction [17]. Nsp12 does not function in isolation; it forms a unique and highly processive complex with its essential cofactors, Nsp7 and Nsp8, which together constitute the minimal core component required for coronavirus RNA synthesis [18]. Due to its pivotal role in the viral life cycle, the absence of host homologs, and its high degree of conservation across *Coronaviridae*, Nsp12 represents a prominent target for the development of broad-spectrum antiviral therapeutics [19]. Beyond its canonical role in replication, viral RdRps, including Nsp12, are increasingly recognized for their immunomodulatory functions. For instance, the hepatitis C virus RdRp modulates the host innate immune response by regulating interferon-β expression [20], and SARS-CoV-2 Nsp12 attenuates type I interferon production by inhibiting the nuclear translocation of interferon regulatory factor 3 (IRF3) [21]. In contrast, the role of PDCoV Nsp12 in virus–host interactions remains largely unexplored. Therefore, elucidating the specific mechanisms by which PDCoV Nsp12 interacts with host proteins to facilitate viral replication and subvert immune responses represents a critical area for future investigation.

During productive infection, numerous viruses subvert the host’s translational machinery to facilitate their own replication. A key component of this machinery is the eukaryotic elongation factor 1 complex (eEF1), which is responsible for the GTP-dependent delivery of aminoacyl-tRNA to the ribosomal A-site, thereby playing an indispensable role in peptide chain elongation [22]. This multi-subunit complex is structurally organized into the GTP-binding subunit eEF1A and the guanine nucleotide exchange factor (GEF) subcomplex eEF1B. The eEF1B subcomplex itself comprises three subunits: eEF1Bα, eEF1Bβ, and eEF1Bγ (also known as eEF1G) [23]. Beyond its canonical role in translation, eEF1G has been implicated in the replication cycles of diverse viruses. For instance, both eEF1A and eEF1G are integral components of the human immunodeficiency virus type 1 reverse transcription complex [24]. Similarly, eEF1G facilitates the viral protein synthesis of Influenza A virus in a strain-specific manner [25]. Furthermore, homologs of eEF1A and eEF1G in plants and yeast have been shown to interact directly with the Tombusvirus RdRp and specific stem-loop structures within the viral RNA genome to promote replication [26]. Despite its established role in the replication of other viruses, the potential function of eEF1G in PDCoV infection remains entirely unexplored. Therefore, this study aims to systematically investigate the biological impact of eEF1G on PDCoV replication and to elucidate the underlying molecular mechanisms. The findings are expected to clarify the precise role of eEF1G in the PDCoV life cycle and may provide a foundational rationale for novel antiviral therapeutic strategies.

## 2. Materials and Methods

### 2.1. Cells and Virus

Lilly Laboratories Culture-Porcine Kidney 1 (LLC-PK1) cells were cultured in Dulbecco’s modified Eagle’s Medium (DMEM; Thermo Fisher Scientific, Waltham, MA, USA) supplemented with 10% fetal bovine serum (FBS; Gibco, Carlsbad, CA, USA), 1% penicillin–streptomycin (Beyotime Biotechnolgy, Shanghai, China), and nonessential amino acids (Gibco). Intestinal porcine epithelial cells (IPEC-J2) cells were cultured in DMEM containing 10% FBS, 1% penicillin–streptomycin, and ITS liquid media supplement (Sigma-Aldrich; St. Louis, MO, USA). BHK-21, swine testis (ST), and MARC-145 cells were cultured in DMEM supplemented with 10% FBS and 1% penicillin–streptomycin. All cells were cultured at 37 °C under 5% CO_2_. The PDCoV strain CHN-HN-1601 (GenBank accession no. MG832584) was propagated and titrated on LLC-PK1 cells using DMEM supplemented with 10 μg/mL trypsin. The PEDV strain BJ2011C (GenBank accession no. KX066126.1) was propagated and titrated on MARC-145 cells using DMEM supplemented with 10 μg/mL trypsin. The PEAV strain GDZQ-2018 (GenBank accession no. MW727454.1) was propagated and titrated on ST cells using DMEM supplemented with 7.5 μg/mL trypsin.

### 2.2. Plasmid Construction

A codon-optimized gene encoding PDCoV Nsp12 was synthesized by Beijing Tsingke Biotech Co., Ltd (Beijing, China). and subsequently cloned into the pCAGGS-HA and p3×FLAG-CMV-10 vectors. Genes for porcine eEF1G, HSPA6, LGALS1, LGALS3, and MX2 were also synthesized and individually inserted into the pCAGGS-HA vector. Truncated mutants of porcine eEF1G and PDCoV Nsp12 were cloned into the pCAGGS-HA and p3×FLAG-CMV-10 vectors, respectively. Other PDCoV nonstructural protein genes were cloned into either the p3×FLAG-CMV-10 or pCAGGS-HA vector. For protein expression, the PDCoV Nsp12 gene was inserted into the pGEX-6P-1 vector to generate a recombinant plasmid expressing glutathione S-transferase (GST)-tagged Nsp12 protein. Similarly, the porcine eEF1G gene was inserted into a pET-28a vector, incorporating an N-terminal Strep II tag, to create the recombinant Strep II-eEF1G plasmid. All primer sequences used for cloning are provided in Appendix A.

### 2.3. Commercial and In-House Generated Antibodies

A suite of commercial and in-house generated antibodies was employed in this study. Commercially sourced antibodies are as follows: anti-Flag tag mouse monoclonal antibody (mAb; M185-3) and anti-His tag mouse mAb (D291-3) were from Medical & Biological Laboratories (MBL; Nagoya, Japan). Anti-HA tag mouse mAb (M180-3) and an additional anti-HA tag rabbit mAb (3724) were procured from MBL and Cell Signaling Technology (Danvers, MA, USA), respectively. Anti-eEF1G rabbit polyclonal antibody (pAb; A7891) was from ABclonal Technology (Wuhan, China), and anti-eEF1G mouse mAb (sc-393378) was from Santa Cruz Biotechnology (Dallas, TX, USA). Anti-GST tag mouse mAb (M20007) was from Abmart (Shanghai, China). Anti-Strep-tag mouse mAb (K200012M) was from Solarbio (Beijing, China). Anti-GAPDH mouse mAb (60004-1-Ig) was from Proteintech (Rosemont, IL, USA). Anti-dsRNA mouse mAb (J2; 10010200) was from Scicons (Budapest, Hungary). Additionally, the following antibodies were generated and validated in our laboratory: rabbit anti-PDCoV M pAb, rabbit anti-PDCoV Nsp12 pAb, mouse anti-PEDV N mAb, and mouse anti-PEAV N mAb.

### 2.4. RNA Extraction and RT-PCR

Total RNA was isolated from mock-infected and PDCoV-infected IPEC-J2 cells using the RNAprep Pure Cell/Bacteria Kit (Tiangen; Beijing, China), in accordance with the manufacturer’s protocol. RNA concentration was quantified using a NanoDrop™ Lite spectrophotometer (Thermo Fisher Scientific). To enable strand-specific detection of negative-sense viral RNA, a reverse transcription primer was designed to anneal to the positive-sense genomic RNA within the PDCoV 5′ UTR (5′-TCATGGTGGCGAATAAGCATTAGCGGCTTGTGGTTT-3′). A primer targeting the porcine β-actin transcript (5′-CTAGAAGCATTTGCGGTGGAC-3′) was used for normalization. For cDNA synthesis, 2 μg of total RNA from each sample was reverse transcribed using the FastKing RT Kit (Tiangen) with integrated genomic DNA removal. RNA fragments corresponding to the 5′ UTR, 3′ UTR, ORF1a, ORF1b, M, and N regions of the PDCoV genome were amplified by 2 × Taq Plus Master Mix (Dye Plus) (Vazyme, Nanjing, China). All primer sequences used are provided in Appendix A.

### 2.5. Real-Time Quantitative PCR (RT-qPCR)

RT-qPCR was performed using Taq Pro Universal SYBR qPCR Master Mix (Vazyme) to quantify target transcript levels. For relative quantification of gene expression, the 2^–ΔΔCT^ method was employed, normalizing cycle threshold (Ct) values to the endogenous β-actin reference gene. To absolutely quantify PDCoV genomic RNA, a standard curve was generated by cloning the PDCoV 5′ UTR region into the pCAGGS-HA vector. Serially diluted plasmids of known concentration were used as templates to establish a linear correlation between Ct values and log10-transformed plasmid copy numbers. Viral RNA copy numbers in experimental samples were subsequently interpolated from this standard curve. All primer sequences used are provided in Appendix A. Prior to analysis, the specificity of each primer pair was confirmed by melt curve analysis, and amplification efficiency was validated to be within the optimal range (90–110%) for reliable quantification.

### 2.6. Western Blot

Following specified treatments, cells were gently washed twice with ice-cold phosphate-buffered saline (PBS) and subsequently lysed on ice using RIPA lysis buffer (Beyotime) supplemented with 1 mM PMSF protease inhibitor. The lysates were clarified by centrifugation at 12,000× *g* for 30 min at 4 °C. Total protein concentration of the supernatants was determined using the Pierce™ BCA Protein Assay Kit (Thermo Fisher Scientific). Equal amounts of protein (20 µg per lane) were resolved by sodium dodecyl sulfate-polyacrylamide gel electrophoresis (SDS-PAGE) and electrophoretically transferred onto polyvinylidene difluoride (PVDF) membranes (Millipore, Bedford, MA, USA). The membranes were blocked for 1 h at room temperature with 5% non-fat dry milk prepared in PBS containing 0.05% Tween-20 (PBST). Subsequently, membranes were probed with the indicated primary antibodies diluted in blocking buffer overnight at 4 °C. Following three washes with PBST, membranes were incubated with species-matched horseradish peroxidase (HRP)-conjugated secondary antibodies for 1 h at room temperature. After another series of three washes with PBST, immunoreactive protein bands were visualized using an enhanced chemiluminescence detection reagent (Engreen; Beijing, China) and imaged with a chemiluminescence imaging system.

### 2.7. Co-Immunoprecipitation (Co-IP)

Following harvesting, cells were lysed and divided into two aliquots. One aliquot was directly analyzed by Western blot to determine total protein expression levels. The second aliquot was subjected to Co-IP by incubation with the specified antibodies overnight at 4 °C. The resulting antibody–antigen complexes were subsequently captured by incubation with Pierce™ Protein A/G Magnetic Beads (Thermo Fisher Scientific) for 2 h at room temperature with gentle rotation. The beads were then washed five times with Tris-buffered saline containing 0.05% Tween-20 (TBST) to remove non-specifically bound proteins. Finally, the immunoprecipitated proteins were eluted from the beads and analyzed by Western blot.

### 2.8. Liquid Chromatography-Tandem Mass Spectrometry (LC-MS/MS)

The pCAGGS-HA-Nsp12 plasmid or the empty pCAGGS-HA vector control was transfected into IPEC-J2 cells using Lipofectamine^®^ LTX Reagent with PLUS™ Reagent (Thermo Fisher Scientific). Immunoprecipitation was subsequently performed using a mouse anti-HA antibody. The immunoprecipitated complexes were separated by SDS-PAGE and analyzed by LC-MS/MS at Bioprofile Technology Company Ltd. (Shanghai, China). To identify Nsp12-specific interacting partners, significantly enriched proteins were screened against the following criteria: (a) unique presence in the pCAGGS-HA-Nsp12 group, and (b) a fold-change ≥ 5 compared to the pCAGGS-HA control group.

### 2.9. RNA Interference (RNAi)

An siRNA targeting eEF1G (si*eEF1G*: 5′-AAUUUACGGAGAAAUUCGG-3′) was designed using the online algorithm DSIR (http://biodev.extra.cea.fr/DSIR/DSIR.html accessed on 26 October 2022). A non-targeting scrambled siRNA (si*NC*: 5′-UUCUCCGAACGUGUCACGU-3′) and the si*eEF1G* duplex were synthesized by GenePharma Co., Ltd. (Shanghai, China). Transfection of the siRNAs was performed into IPEC-J2 cells using Lipofectamine™ 2000 Reagent (Thermo Fisher Scientific) in accordance with the manufacturer’s protocol. Knockdown efficiency was assessed at 36 h post-transfection by Western blot analysis using a rabbit anti-eEF1G antibody, with GAPDH serving as a loading control.

### 2.10. 50% Tissue Culture Infectious Dose (TCID_50_) Assay

Viral yields were quantified by determining the TCID_50_ on LLC-PK1 cells. Briefly, viral samples were serially diluted ten-fold in serum-free DMEM supplemented with 10 μg/mL trypsin. The diluted samples were then inoculated onto LLC-PK1 cell monolayers in 96-well plates. At 48 h post-infection (hpi), infection was assessed by indirect immunofluorescence assay (IFA). The final viral titers were calculated using the Reed-Muench method.

### 2.11. IFA

Cells were fixed with 4% paraformaldehyde for 10 min at room temperature, permeabilized with 0.2% Triton X-100 in PBS for 10 min, and then blocked with 2% bovine serum albumin (BSA) in PBS for 30 min. After three washes with PBS, the cells were incubated with the specified primary antibodies for 1 h at 37 °C. Following another three washes with PBS, the cells were incubated with species-appropriate fluorescently conjugated secondary antibodies for 1 h at 37 °C. Nuclei were counterstained with 4′,6-diamidino-2-phenylindole (DAPI; Thermo Fisher Scientific) for 5 min and washed three times with PBS. Images were acquired using a Nikon TS2 microscope (Nikon, Tokyo, Japan) and analyzed with NIS-Elements Viewer software (version 5.21).

### 2.12. RNA Immunoprecipitation (RIP)

LLC-PK1 cells in 10 cm dishes were infected with PDCoV at a multiplicity of infection (MOI) of 1 for 12 h, then gently washed with PBS. Cells were lysed on ice using RIPA lysis buffer supplemented with 1 mM PMSF and an RNase Inhibitor (MedChemExpress; Monmouth Junction, NJ, USA). The lysates were clarified by centrifugation at 12,000× *g* for 30 min at 4 °C. The supernatant was incubated overnight at 4 °C with 5 µg of either anti-eEF1G rabbit pAb or normal rabbit IgG (negative control). The antibody–protein complexes were subsequently captured by incubation with 25 µL of Pierce™ Protein A/G Magnetic Beads (Thermo Fisher Scientific) for 2 h at room temperature. Beads were washed five times with RIP lysis buffer and then collected for parallel analysis by Western blot and RNA extraction. Associated RNA was detected by RT-PCR. For the reciprocal RNA-protein binding assay, purified recombinant proteins and viral RNA were incubated overnight at 4 °C in different orders. The resulting complexes were immunoprecipitated with an anti-PDCoV Nsp12 pAb (with normal rabbit IgG as a negative control) overnight at 4 °C, followed by capture with Protein A/G Magnetic Beads for 2 h at room temperature. After five washes, the beads were processed for Western blot analysis and RNA extraction. Co-precipitated RNA was quantified by RT-qPCR.

### 2.13. Strep II Pull-Down Assay

Recombinant plasmids pET-28a-Strep II-eEF1G and pGEX-6P-1-Nsp12 were individually transformed into *E. coli* BL21(DE3) competent cells. Protein expression was induced with 1 mM isopropyl β-D-1-thiogalactopyranoside (IPTG) for 20 h at 16 °C. Bacterial cultures were harvested by centrifugation, resuspended in PBS, and lysed by sonication. The lysates were clarified by centrifugation, and the supernatants containing GST-Nsp12 fusion protein were incubated with Glutathione Sepharose 4B beads (Cytiva Lifesciences; Marlborough, MA, USA), while those containing Strep II-eEF1G were incubated with Streptactin Beads 4FF (Smart-Lifesciences; Changzhou, China) overnight at 4 °C. After five washes with PBS, the recombinant proteins were eluted and analyzed by SDS-PAGE to assess purity. For the pull-down assay, 10 μg of purified Strep II-eEF1G was immobilized onto 50 μL of Streptactin Beads 4FF overnight at 4 °C. After washing the beads five times with PBS, 20 μg of purified GST-Nsp12 was added and incubated at room temperature for 2 h. Following five additional washes with PBS, the bound protein complexes were eluted and analyzed by Western blot. Purified GST protein served as a negative control.

### 2.14. RNAscope In Situ Hybridization

In situ hybridization was performed to detect both positive- and negative-stranded PDCoV genomic RNA in infected IPEC-J2 cells. The assay utilized custom-designed RNAscope probes (Advanced Cell Diagnostics; Newark, CA, USA) targeting the 5′ UTR of the positive strand and the N gene of the negative strand of the PDCoV strain CHN-HN-1601. The assay was carried out using the RNAscope Multiplex Fluorescent Detection Reagents v2 Kit (Advanced Cell Diagnostics) in strict accordance with the manufacturer’s protocol. The positive- and negative-stranded PDCoV RNA were visualized using the TSA Vivid Fluorophore Kit 650 (Bio-Techne; Lille, France).

### 2.15. Viral Adsorption and Internalization Assay

IPEC-J2 cells were transfected with the indicated siRNA for 36 h. For the viral adsorption assay, the transfected cells were then prechilled at 4 °C for 30 min, followed by infection with PDCoV at an MOI of 1 and incubated at 4 °C for an additional 1 h to allow virus adsorption but impede its internalization. Cells were washed three times with prechilled PBS to remove the unbound virions. For the viral internalization assay, after completing the viral adsorption assay, the cells were incubated at 37 °C for another 1 h to allow viral internalization. Cells were washed three times for 1 min with citric acid buffer (citric acid 40 mM, potassium chloride 10 mM, sodium chloride 135 mM; pH 3.0) to remove non-internalized virions from the cell surface. The cells were subjected to RNA extraction and RT-qPCR.

### 2.16. Cell Viability Assay

Cell viability was evaluated using a Cell Counting Kit-8 (CCK-8; MedChemExpress) assay. At the specified time points, 10 μL of CCK-8 reagent was added to each well. The plates were subsequently incubated at 37 °C for 4 h, after which the absorbance was measured at a wavelength of 450 nm using a microplate spectrophotometer.

### 2.17. Statistical Analysis

All statistical analyses were performed using GraphPad Prism software (version 8.0). Data obtained from a minimum of three independent experiments are presented as mean ± standard deviation (SD). Comparisons between two groups were analyzed using an unpaired Student’s *t*-test. For comparisons among three or more groups, one-way analysis of variance (ANOVA) or two-way ANOVA was employed, as appropriate. Statistical significance is denoted with asterisks: * *p* < 0.05, ** *p* < 0.01, and *** *p* < 0.001; NS indicates not significant. Error bars on graphical data represent SD.

## 3. Results

### 3.1. Screening and Identification of Host Proteins Interacting with PDCoV Nsp12 Protein

Coronavirus Nsp12, which possesses RdRp activity, is essential for the assembly of the RTC. Previous studies have shown that SARS-CoV Nsp12 forms the RTC by recruiting Nsp7 and Nsp8, thereby exhibiting replicase activity [27]. Given its critical role in viral replication, we used an immunoprecipitation-coupled mass spectrometry approach to identify host proteins that interact with PDCoV Nsp12. We identified 94 differentially expressed proteins (DEPs) and performed functional annotation using Gene Ontology (GO) and Kyoto Encyclopedia of Genes and Genomes (KEGG) pathway analyses. Among these, five DEPs—eEF1G [24], LGALS1, LGALS3 [28], HSPA6 [29], and MX2 [30]—have previously been linked to viral replication and were selected for further investigation. To test their interactions with PDCoV Nsp12, BHK-21 cells were co-transfected with a plasmid encoding Nsp12 and one encoding each of the five host proteins. Co-IP analysis of total cellular proteins revealed that eEF1G, LGALS1, and HSPA6 interact with PDCoV Nsp12, while LGALS3 and MX2 do not (Figure 1).

### 3.2. Cellular eEF1G Directly Interacts with the PDCoV Nsp12 Protein

Of the three host proteins (eEF1G, LGALS1, and HSPA6) found to interact with the PDCoV Nsp12 protein, eEF1G was chosen for further investigation. eEF1G is a subunit of eEF1, which plays a critical role in peptide chain elongation during eukaryotic protein synthesis [22]. Having confirmed the interaction between eEF1G and Nsp12 under transfection conditions (Figure 1), we next examined whether this interaction also occurs during actual PDCoV infection. To do so, both IPEC-J2 and LLC-PK1 cells were either mock-infected or infected with PDCoV and subjected to Co-IP analysis using a rabbit anti-Nsp12 pAb. The results revealed that eEF1G was specifically immunoprecipitated by the anti-Nsp12 antibody, but not by the isotype control (IgG), in both PDCoV-infected cell lines (Figure 2A). Confocal immunofluorescence analysis further demonstrated partial co-localization of Nsp12 and eEF1G in the cytoplasm of PDCoV-infected IPEC-J2 cells (Figure 2B and Appendix A). Selection of time points was guided by both the established PDCoV replication cycle, which is completed in about 5–6 h [31], and our own characterization of the CHN-HN-1601 strain (Appendix A). Importantly, in vitro pull-down assays using prokaryotically expressed GST-Nsp12 and Strep II-eEF1G proteins confirmed that GST-Nsp12 was pulled down by Strep II-eEF1G, but not by the GST tag alone (Figure 2C,D), indicating a direct interaction between eEF1G and PDCoV Nsp12. Collectively, these findings demonstrate that eEF1G interacts directly with PDCoV Nsp12. Additionally, Co-IP assays were also performed to test interactions between eEF1G and other PDCoV Nsps. The results showed that eEF1G specifically interacts only with Nsp12, but not with other Nsps (Appendix A).

Next, we constructed a series of protein truncations to identify the key regions responsible for the interaction between eEF1G and Nsp12 (Figure 2E,F). The truncations were designed primarily based on the functional domain boundaries of eEF1G and Nsp12. Specifically, eEF1G contains three functional domains: GST N-terminal, GST C-terminal, and EF-1-gamma C-terminal. Similarly, Nsp12 of PDCoV, like that of other CoVs, also comprises three domains: the NiRAN, interface, and RdRp domains [17,18]. The plasmid encoding FLAG-tagged Nsp12 was co-transfected into BHK-21 cells along with recombinant plasmids expressing either full-length eEF1G or one of its six truncated constructs. Cell lysates were then analyzed by Co-IP. As shown in Figure 2G, only constructs containing the aa348–378 region coimmunoprecipitated with Nsp12, suggesting that this segment within the EF-1-gamma C-terminal domain of eEF1G mediates the interaction. Similarly, five truncation mutants of Nsp12 were generated and tested for binding to eEF1G. Full-length Nsp12, aa218–781, aa1–665, aa1–529, and aa1–432 all coimmunoprecipitated with eEF1G, while aa1–218 did not (Figure 2H), indicating that the aa219–432 region within the RdRp domain is essential for the interaction with eEF1G. In addition, we employed the ZDOCK online server (https://zdock.wenglab.org/ accessed on 25 March 2025) to predict potential interacting residues between the eEF1G segment (aa348–378) and the Nsp12 region (aa219–432). The top-ranked model suggests that hydrogen bonds mediate the interaction between the two proteins. Specifically, threonine (Thr360), asparagine (Asn361), and serine (Ser363) in eEF1G form hydrogen bonds with arginine (Arg290), tyrosine (Tyr294), and isoleucine (Ile289) in Nsp12, respectively (Figure 2I).

### 3.3. Cellular eEF1G Acts as a Host Restriction Factor for PDCoV

To investigate the role of eEF1G in PDCoV replication, eEF1G-specific siRNA was designed to knockdown endogenous eEF1G expression in IPEC-J2 cells, followed by assessment of viral replication. The results demonstrated a significant increase in PDCoV M protein levels in si*eEF1G*-transfected cells compared to those transfected with scrambled control siRNA (si*NC*), starting from 12 hpi (Figure 3A). IFA further revealed that PDCoV replication was significantly enhanced in eEF1G-knockdown IPEC-J2 cells compared to si*NC*-transfected cells, as evidenced by a significant increase in the number of cells showing positive immunofluorescence staining for the PDCoV M protein (Figure 3B,C). Viral titer assays further confirmed that knockdown of eEF1G significantly increased the production of PDCoV progeny viruses in IPEC-J2 cells relative to si*NC*-transfected controls (Figure 3D). We have investigated the role of eEF1G in the replication of two additional coronaviruses: PEDV and PEAV. Immunofluorescence assays revealed that the replication of both viruses was significantly enhanced in eEF1G-knockdown cells compared to the si*NC* control (Appendix A). This was evidenced by a marked increase in the number of cells positive for the PEAV N and PEDV N proteins, respectively (Appendix A). To rule out the possibility that the increased replication capacity of PDCoV was due to the effect of si*eEF1G* transfection on the viability of IPEC-J2 cells, we assessed the impact of si*eEF1G* transfection on IPEC-J2 cell viability using the CCK-8 assay. The results showed no significant difference in cell viability between cells transfected with si*eEF1G* and si*NC* (Figure 3E). Furthermore, the impact of eEF1G overexpression on PDCoV replication was also evaluated. IPEC-J2 cells transfected with eEF1G-expressing plasmids were infected with PDCoV and then analyzed by Western blot, viral TCID_50_ assay, and IFA. The results demonstrated no significant differences in PDCoV M protein expression, viral titers, or the proportion of PDCoV M-positive cells between the eEF1G overexpression group and the empty vector control (Appendix A). Taken Together, these findings indicate that knockdown of eEF1G promotes PDCoV replication in IPEC-J2 cells, whereas overexpression of eEF1G does not affect viral growth, suggesting that eEF1G acts as a host restriction factor for PDCoV.

### 3.4. eEF1G Functions to Inhibit the Synthesis of PDCoV Negative-Stranded RNA

Although eEF1G is a translation factor localized to both the cytoplasm and nucleus, our initial experiments ruled out its role in the early stages of viral entry. Specifically, the knockdown of eEF1G resulted in no significant differences in viral adsorption or internalization compared to the negative control (Figure 4A,B). To investigate its potential role in PDCoV RNA replication, we performed a strand-specific RT-qPCR assay. The results demonstrated that the level of negative-stranded RNA was significantly increased in the si*eEF1G* group compared to the si*NC* group at 9 (*p* < 0.001), 12 (*p* < 0.01), and 15 hpi (*p* < 0.001) (Figure 4C). In contrast, no significant differences in the levels of positive-stranded RNA or subgenomic mRNA were observed between the two groups (Figure 4D,E). These data indicate that eEF1G functions to inhibit the synthesis of PDCoV negative-stranded RNA during infection.

### 3.5. eEF1G Binds to the Genomic RNA of PDCoV During Infection

To elucidate the mechanism by which eEF1G inhibits PDCoV replication, we first assessed whether its expression or localization is altered upon infection. RT-qPCR analysis (using β-actin for normalization) confirmed that eEF1G mRNA levels remain stable throughout infection (Appendix A). Western blot analysis revealed no significant difference in eEF1G protein levels between PDCoV-infected and mock-infected IPEC-J2 and LLC-PK1 cells (Appendix A). Similarly, transfection of IPEC-J2 cells with an Nsp12-expressing plasmid did not affect eEF1G abundance (Appendix A). Furthermore, IFA showed that eEF1G exhibits a diffuse localization pattern in both the cytoplasm and nucleus, which was unchanged by PDCoV infection (Appendix A). Collectively, these results demonstrate that PDCoV infection does not affect the abundance or subcellular localization of eEF1G.

Having demonstrated that eEF1G inhibits negative-stranded RNA synthesis of PDCoV, we hypothesized that eEF1G may bind to viral genomic RNA, thereby suppressing genomic RNA amplification. To test this, an RIP assay was performed to examine the potential interaction between eEF1G and PDCoV RNA. In both PDCoV-infected IPEC-J2 and LLC-PK1 cells, RNA fragments corresponding to the 5′ UTR, 3′ UTR, ORF1a, ORF1b, M, and N regions of the PDCoV genome were enriched in eEF1G immunoprecipitates. In contrast, GAPDH mRNA, used as a control, was not enriched (Figure 5A,B). These results indicate that eEF1G specifically binds to PDCoV genomic RNA during viral infection.

### 3.6. Cellular eEF1G Disrupts the Binding of Nsp12 to the PDCoV Genomic RNA During Infection

Building upon established evidence that the host factor eEF1G directly interacts with PDCoV Nsp12 and binds to PDCoV genomic RNA during infection, we further investigated the interplay among eEF1G, Nsp12, and PDCoV genomic RNA using confocal immunofluorescence assays combined with RNAscope in situ hybridization. Results demonstrated that both eEF1G and Nsp12 partially co-localize with positive- and negative-stranded viral RNA (Figure 6A), suggesting their potential interaction during PDCoV infection. Subsequent three-dimensional reconstruction analysis confirmed that a subset of Nsp12 and eEF1G co-localizes with PDCoV genomic RNA (Appendix A), providing further evidence for their association within infected cells.

A canonical function of the RdRp is to bind RNA and nucleotides [32]. To determine if PDCoV Nsp12 possesses this RNA-binding activity, we performed a solution-based binding assay. Purified GST-Nsp12 was incubated with total RNA from PDCoV-infected IPEC-J2 cells, and the complexes were immunoprecipitated with an anti-Nsp12 antibody. RT-qPCR analysis of the PDCoV 5′ UTR revealed a specific enrichment of viral RNA in the Nsp12 pulldown compared to the control IgG (Figure 6B). Interestingly, this enrichment decreased dose-dependently with the addition of Strep II-eEF1G, suggesting eEF1G may compete for RNA binding. To test this hypothesis, we investigated whether eEF1G binding to the PDCoV genomic RNA impedes its association with Nsp12. Purified GST-Nsp12, Strep II-eEF1G, and PDCoV RNA were co-incubated and immunoprecipitated with an anti-Nsp12 antibody. Absolute RT-qPCR quantification showed that eEF1G co-incubation significantly reduced the amount of PDCoV RNA bound to Nsp12 by up to 549-fold (*p* < 0.001; Figure 6B). To elucidate the mechanism of interference, we analyzed the binding hierarchy. We pre-incubated purified Strep II-eEF1G with either GST-Nsp12 or PDCoV RNA before adding the remaining component. The most pronounced reduction in Nsp12-RNA binding (a 171-fold decrease, *p* < 0.001) occurred when eEF1G was pre-incubated with the viral RNA (Figure 6C). This indicates that eEF1G binding to the genomic RNA sterically hinders subsequent recruitment of Nsp12. We next asked if this competition occurs in a cellular context during infection. PDCoV-infected IPEC-J2 cells, pre-treated with siRNAs, were subjected to immunoprecipitation with an anti-Nsp12 antibody. The amount of co-precipitated viral RNA was significantly higher in eEF1G-knockdown (si*eEF1G*) cells than in control (si*NC*) cells (*p* < 0.001; Figure 6D). Collectively, these data demonstrate that eEF1G hinders the interaction between Nsp12 and the PDCoV genomic RNA by preferentially binding to the viral RNA. This competition provides a mechanism by which eEF1G may inhibit the synthesis of negative-stranded PDCoV RNA by obstructing the initial binding of the viral RdRp to its genomic template.

## 4. Discussion

The Nsps of CoVs constitute the fundamental machinery for viral replication and transcription. This includes the formation of double-membrane vesicles, which serve as protected replication organelles, and the assembly of the multi-protein RTC [33,34]. Among these, the RdRp, Nsp12, functions as the catalytic core of the RTC. While the biochemical properties of coronavirus Nsp12 are well-characterized, significant knowledge gaps remain regarding its specific interactions with host proteins, particularly in the case of PDCoV. Elucidating these virus–host interactions is critical for understanding the molecular mechanisms of viral pathogenesis and may reveal pan-coronaviral therapeutic targets. In this study, we identify eEF1G as a novel host interactor of PDCoV Nsp12 and demonstrate its potent inhibitory effect on viral propagation. We further delineate the mechanism: eEF1G directly binds to both Nsp12 and the viral genomic RNA. This dual interaction competitively hinders the formation of the essential Nsp12–genomic RNA complex, consequently impairing the synthesis of negative-stranded RNA, a critical replicative intermediate. Based on these findings, we propose a model detailing the molecular mechanism by which eEF1G restricts PDCoV replication (Figure 7).

Given the central role of Nsp12 in viral genome replication and transcription, hosts have evolved various defense mechanisms to counteract its function. For instance, transmembrane protein 53 has been shown to interact with PEAV Nsp12 and disrupt the assembly of the viral RdRp complex, thereby suppressing viral RNA synthesis [35]. Similarly, Alpha-actinin-4 was identified as a binding partner of SARS-CoV-2 Nsp12, competing with viral RNA for access to the RdRp complex and consequently inhibiting viral replication [36]. Owing to a high degree of sequence and structural conservation of Nsp12 across *Coronaviridae*, insights derived from other CoVs are highly relevant to the study of PDCoV [37]. Building on previous work that employed immunoprecipitation coupled with mass spectrometry to identify host proteins interacting with SARS-CoV-2 Nsp12 [38], we adopted an analogous proteomic approach to screen for PDCoV Nsp12-binding partners. Our analysis revealed that the DEPs are predominantly associated with key cellular processes such as apoptosis, autophagy, energy metabolism, and innate immunity. These findings suggest that PDCoV Nsp12, much like its SARS-CoV-2 counterpart, may play multifaceted roles in modulating host cell processes—including cell proliferation, cell death, and immune regulation [21]. Further investigation is warranted to elucidate the precise mechanisms by which Nsp12 influences these pathways during PDCoV infection.

The eEF1 is a central component of the host translational machinery, responsible for peptide chain elongation during protein synthesis [22]. It consists of two core subunits: eEF1A, which binds GTP, and eEF1B, a multimeric GEF composed of eEF1Bα, eEF1Bγ (eEF1G), and eEF1Bδ (eEF1D). Within this complex, eEF1G is thought to act as a structural scaffold essential for eEF1B assembly [23]. A study demonstrated that eEF1Bβ interacts with eEF1G with high functional affinity, suggesting that eEF1Bβ may modulate the formation of an ordered eEF1G oligomer, a structure potentially critical for protein biosynthesis [39]. However, the upstream pathways that regulate eEF1G’s function remain poorly characterized. Beyond its role in the complex, eEF1G function can also be modulated by post-translational modification. For instance, its C-terminus contains a conserved LAMMER kinase phosphorylation site that is important for its normal function [40]. Beyond its canonical role, eEF1 subunits are frequently co-opted by various viruses to promote their replication. Contrary to this established proviral role for other eEF1 components, our study provides several key advances in the context of coronavirus infection. We demonstrated the first evidence that eEF1G interacts with PDCoV Nsp12 and functions as a potent restriction factor against viral growth. Using a multifaceted approach (Western blot, immunofluorescence, viral titer assays), we demonstrated that siRNA-mediated knockdown of eEF1G significantly enhanced PDCoV replication in IPEC-J2 cells (Figure 3), indicating an intrinsic inhibitory role for eEF1G. The consistent findings across PDCoV, PEDV, and PEAV suggest that eEF1G may inhibit viral replication through a conserved mechanism common to these coronaviruses (Appendix A). Subsequent attempts to generate eEF1G knockout IPEC-J2 cell lines using CRISPR-Cas9 technology were unsuccessful, suggesting that eEF1G is essential for cell viability. Interestingly, and in contrast to the knockdown phenotype, overexpression of eEF1G did not further suppress PDCoV proliferation (Appendix A). We hypothesize that the high constitutive expression of eEF1G, evidenced by its abundant distribution in both the cytoplasm and nucleus (Figure 2B), may already be saturating its antiviral function at baseline levels, rendering additional overexpression functionally redundant. Collectively, these findings unveil a novel and unexpected role for eEF1G as a host restriction factor that impedes PDCoV replication.

Several studies have implicated eEF1G in various stages of viral replication, like transcription and translation [24,25,26]. As coronaviruses enter cells via receptor-mediated endocytosis and begin replication upon the release of genomic RNA into the cytoplasm, we re-evaluated the potential role of eEF1G in these early stages of PDCoV infection. Our results confirm that eEF1G does not influence the initial phases of the viral lifecycle. Specifically, RT-qPCR analyses showed no significant effect of eEF1G knockdown on viral adsorption or internalization (Figure 4A,B). Furthermore, eEF1G knockdown did not alter the expression level of the Nsp12 protein (Figure 6D), indicating it does not affect the translation of viral proteins. Our results indicate that eEF1G specifically inhibits the synthesis of negative-stranded RNA, but not positive-stranded or subgenomic RNAs, during PDCoV infection (Figure 4). This contrasts with the proviral roles described in other systems and highlights that eEF1G can regulate viral RNA synthesis through distinct, virus-specific mechanisms at different phases of the replication cycle. As host proteins are often recruited to viral replication complexes to either facilitate or restrict viral replication [41], we proposed that eEF1G might be translocated to these sites to exert its inhibitory function. However, immunofluorescence analysis revealed that the subcellular distribution of eEF1G remains unchanged in PDCoV-infected IPEC-J2 cells, with no observable re-localization to viral replication complexes (Appendix A). Furthermore, RT-qPCR and Western blot analysis confirmed that eEF1G protein levels remain stable upon PDCoV infection in both IPEC-J2 and LLC-PK1 cells (Appendix A), indicating that the virus does not modulate its abundance. Based on reports that certain flavonoids inhibit hepatitis C virus RdRp activity by competing for RNA binding [42], and given that eEF1G interacts with PDCoV Nsp12 and selectively impedes negative-stranded RNA synthesis, we speculated that eEF1G may disrupt the RNA-binding function of Nsp12. RIP assays confirmed that eEF1G binds PDCoV genomic RNA in infected cells (Figure 5). Since template RNA binding by Nsp12 is essential for coronavirus replication [43,44], we hypothesized that eEF1G may compete with Nsp12 for genomic RNA access, thereby inhibiting negative-stranded synthesis. This was supported by confocal imaging showing co-localization among eEF1G, Nsp12, and viral RNA in infected cells (Figure 6A). In vitro competitive binding assays demonstrated that pre-incubation of PDCoV RNA with eEF1G significantly reduces the amount of RNA bound by Nsp12 (Figure 6B,C). A similar effect was observed in PDCoV-infected cells (Figure 6D). These findings suggest two non-mutually exclusive mechanisms: (a) eEF1G exhibits higher affinity for PDCoV RNA than Nsp12, or (b) eEF1G binding sterically hinders Nsp12–RNA interaction by occluding the polymerase binding site. The order-dependent reduction in Nsp12 binding upon eEF1G pre-incubation supports a model in which eEF1G sequesters the genomic RNA, limiting its availability for Nsp12 (Figure 6C). Further structural studies are needed to elucidate the precise molecular interactions within the Nsp12–eEF1G–RNA complex.

## 5. Conclusions

To our knowledge, this is the first study to demonstrate that the host factor eEF1G interacts directly with PDCoV Nsp12 and binds viral genomic RNA, thereby disrupting the formation of the Nsp12–RNA complex. This interference impairs the synthesis of negative-stranded RNA and ultimately inhibits PDCoV replication. Our findings provide novel insights into the role of host translational machinery in modulating coronavirus infection.

## Figures and Tables

**Figure 1 viruses-17-01369-f001:**
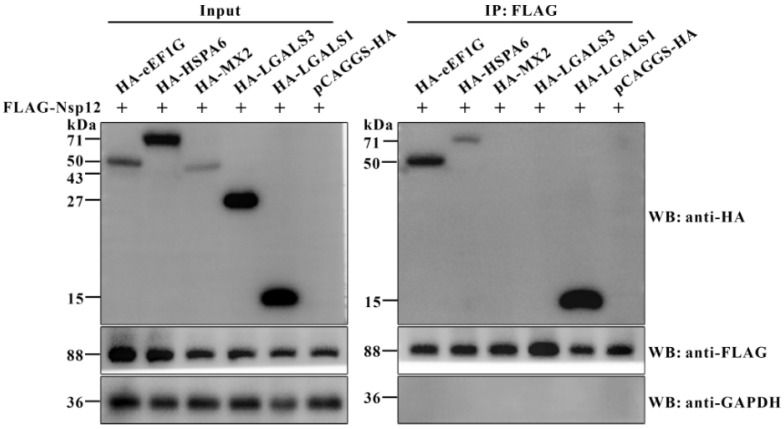
Co-IP analysis of host proteins interacting with PDCoV Nsp12 protein. BHK-21 cells were co-transfected for 36 h with the recombinant plasmid p3×FLAG-CMV-10-Nsp12 (1 μg per well in a six-well plate) along with a pCAGGS-HA plasmid (1 μg per well in a six-well plate) encoding one of the five host proteins (eEF1G, HSPA6, MX2, LGALS3, or LGALS1). Total cellular proteins were subsequently extracted and subjected to Co-IP using an anti-FLAG antibody, with the pCAGGS-HA empty plasmid serving as a control. Precipitated proteins were then analyzed by Western blot with rabbit anti-HA, anti-FLAG, and anti-GAPDH antibodies. “+” represents the presence of the indicated plasmid.

**Figure 2 viruses-17-01369-f002:**
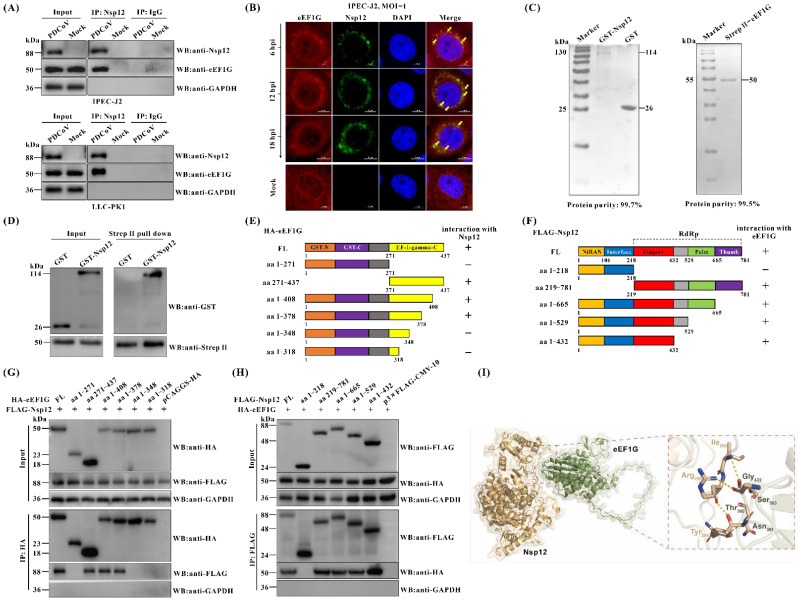
Cellular eEF1G directly interacts with the PDCoV Nsp12 protein. (**A**) IPEC-J2 and LLC-PK1 cells were either mock-infected or infected with PDCoV at an MOI of 1 for 12 h, followed by harvesting for Co-IP assays using a rabbit anti-Nsp12 pAb; normal rabbit IgG was used as a control. Immune complexes were precipitated and analyzed by Western blot with antibodies against PDCoV Nsp12, eEF1G, and GAPDH (rabbit anti-eEF1G antibody was used). (**B**) IPEC-J2 cells were either mock-infected or infected with PDCoV at an MOI of 1. At 6, 12, and 18 hpi, the cells were fixed and processed for confocal immunofluorescence analysis. Following incubation with primary antibodies against Nsp12 and eEF1G (mouse anti-eEF1G antibody was used), the cells were treated with Alexa Fluor 488-conjugated goat anti-rabbit IgG and Alexa Fluor 568-conjugated goat anti-mouse IgG secondary antibodies, respectively. Nuclei were counterstained with DAPI. Pictures represent PDCoV Nsp12 protein (Green), eEF1G (Red), nuclei (Blue), and merged images (Merge). Sites of colocalization are indicated by yellow arrows. Scale bar: 5 μm. (**C**) Recombinant plasmids pET-28a-Strep II-eEF1G and pGEX-6P-1-Nsp12 were transformed into *E. coli* BL21(DE3) cells. Following induction with 1 mM IPTG at 16 °C for 20 h, the Strep II-eEF1G and GST-Nsp12 fusion proteins were purified from the soluble lysate using Streptactin and Glutathione Sepharose beads, respectively. Purified recombinant proteins were analyzed by SDS-PAGE and Coomassie blue staining, and their purity was assessed by quantifying the stained bands via grayscale scanning. (**D**) Ten micrograms of purified Strep II-eEF1G were bound to Streptactin Beads 4FF and subsequently incubated with 20 μg of either purified GST-Nsp12 or the GST tag alone. The resulting complexes were then analyzed by Western blot. (**E**) A schematic diagram of the eEF1G domain structure and the strategy for constructing HA-tagged full-length (FL) and truncated mutants. Grey represents a disordered region. “+” represents interactions and “−” represents negative interactions. (**F**) A schematic diagram of the Nsp 12 domain structure and the strategy for constructing FLAG-tagged full-length (FL) and truncated mutants. Grey represents a disordered region. “+” represents interactions and “−” represents negative interactions. (**G**) BHK-21 cells were co-transfected for 36 h with the p3×FLAG-CMV-10-Nsp12 plasmid (1 μg per well in a six-well plate) and pCAGGS-HA plasmids (1 μg per well in a six-well plate) expressing either full-length eEF1G or a truncated mutant. Cell lysates were then subjected to Co-IP with a mouse anti-HA antibody. Precipitated proteins were then analyzed by Western blot with rabbit anti-HA, anti-FLAG, and anti-GAPDH antibodies. “+” represents the presence of the indicated plasmid. (**H**) BHK-21 cells were co-transfected for 36 h with the pCAGGS-HA-eEF1G plasmid (1 μg per well in a six-well plate) and p3×FLAG-CMV-10 plasmids (1 μg per well in a six-well plate) expressing either full-length Nsp12 or a truncated mutant. Cell lysates were then subjected to Co-IP with an anti-FLAG antibody. Precipitated proteins were then analyzed by Western blot with rabbit anti-HA, anti-FLAG, and anti-GAPDH antibodies. “+” represents the presence of the indicated plasmid. (**I**) The optimal model of the eEF1G–Nsp12 complex, predicted by the ZDOCK server, is depicted with Nsp12 in yellow and eEF1G in green. Interacting residues are highlighted within the dotted box and labeled in corresponding colors. Hydrogen bonds are indicated by yellow dashed lines.

**Figure 3 viruses-17-01369-f003:**
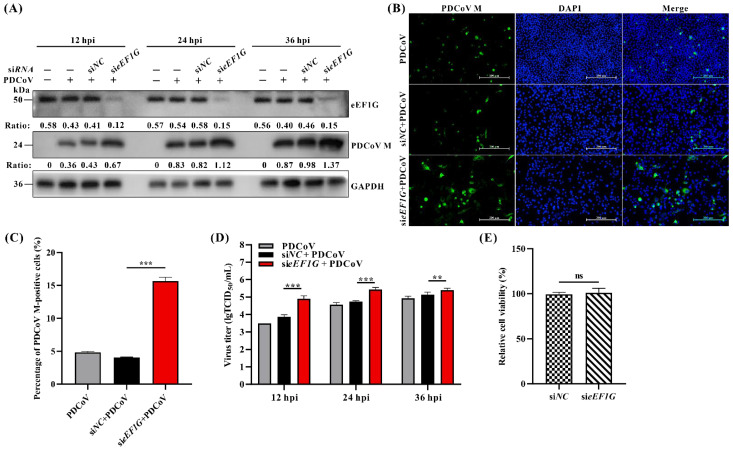
Knockdown of eEF1G significantly promotes PDCoV replication. (**A**) IPEC-J2 cells were transfected with either si*eEF1G* or scrambled control siRNA (si*NC*) for 36 h and then infected with PDCoV at an MOI of 1. The cells were harvested at 12, 24, and 36 hpi for Western blot analysis using primary antibodies specific to PDCoV M protein, eEF1G, and GAPDH (rabbit anti-eEF1G antibody was used). The ratio of each target protein band to its corresponding GAPDH internal reference band is indicated directly below the respective band. “+” represents the presence and “−” represents the absence of indicated PDCoV or siRNA. (**B**) IPEC-J2 cells were transfected with either si*eEF1G* or scrambled control siRNA (si*NC*) for 36 h and then infected with PDCoV at an MOI of 1. At 12 hpi, the cells were fixed and processed for confocal immunofluorescence analysis. After incubation with a primary antibody against PDCoV M protein, the cells were treated with an Alexa Fluor 488-conjugated goat anti-rabbit IgG secondary antibody. Nuclei were counterstained with DAPI. Pictures represent PDCoV M protein (Green), nuclei (Blue), and merged images (Merge). Scale bar: 200 μm. (**C**) Statistical analysis of the percentage of M protein-positive cells among PDCoV-infected IPEC-J2 cells shown in (**B**). *** *p* < 0.001. (**D**) IPEC-J2 cells were transfected with either si*eEF1G* or si*NC* for 36 h and subsequently infected with PDCoV at an MOI of 1. Viral yields were determined by TCID_50_ assay at 12, 24, and 36 hpi. Data, presented as means ± SD from three independent experiments, were analyzed by two-way ANOVA. ** *p* < 0.01, *** *p* < 0.001. (**E**) Cell proliferation activity following siRNA interference was assessed using the CCK-8 assay. Data are presented as mean ± SD from three independent experiments. ns, no significance.

**Figure 4 viruses-17-01369-f004:**
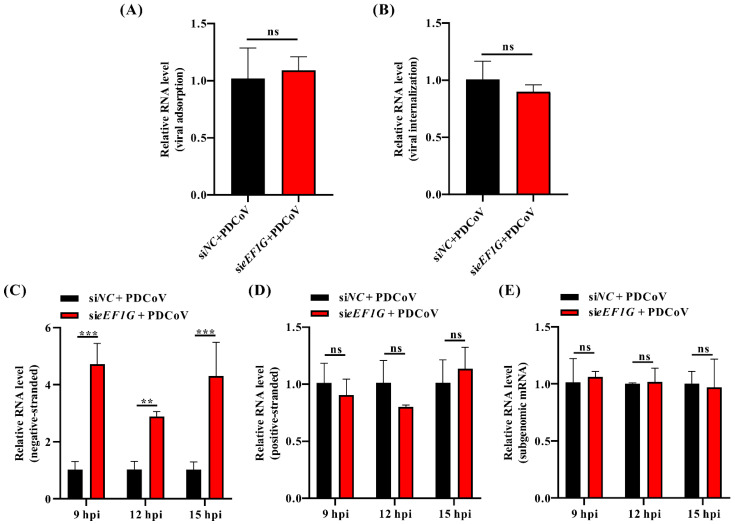
Cellular eEF1G inhibits the synthesis of PDCoV negative-stranded RNA. (**A**) Viral Adsorption and (**B**) Internalization Assays. IPEC-J2 cells were transfected with either si*eEF1G* or si*NC* siRNAs for 36 h. For both assays, cells were prechilled at 4 °C for 30 min and then infected with PDCoV (MOI of 1) at 4 °C for 1 h. Unbound virions were removed by washing three times with prechilled PBS. For the adsorption assay (**A**), viral RNA was quantified immediately after washing. For the internalization assay (**B**), cells were further incubated at 37 °C for 1 h to allow entry, followed by washing with a pH 3.0 citric acid buffer to inactivate non-internalized virus. In both assays, the RNA levels of the PDCoV N gene were quantified by RT-qPCR using β-actin as a reference, and relative levels were normalized to the si*NC* group. Data are presented as the mean ± SD from three independent experiments. ns, no significance. (**C**) IPEC-J2 cells were transfected with either si*eEF1G* or si*NC* siRNAs for 36 h, then infected with PDCoV (MOI = 1). Cells were harvested at 9, 12, and 15 hpi, and PDCoV negative-strand RNA levels were quantified by strand-specific RT-qPCR using β-actin as a reference gene. Relative RNA levels at each time point were normalized to the si*NC* control. Data, presented as means ± SD from three independent experiments, were analyzed by two-way ANOVA. ns, no significance; ** *p* < 0.01; *** *p* < 0.001. (**D**) PDCoV positive-strand RNA levels in IPEC-J2 cells were assessed following transfection, infection, and processing as described in (**C**). (**E**) PDCoV subgenomic mRNA levels in IPEC-J2 cells were assessed following transfection, infection, and processing as described in (**C**).

**Figure 5 viruses-17-01369-f005:**
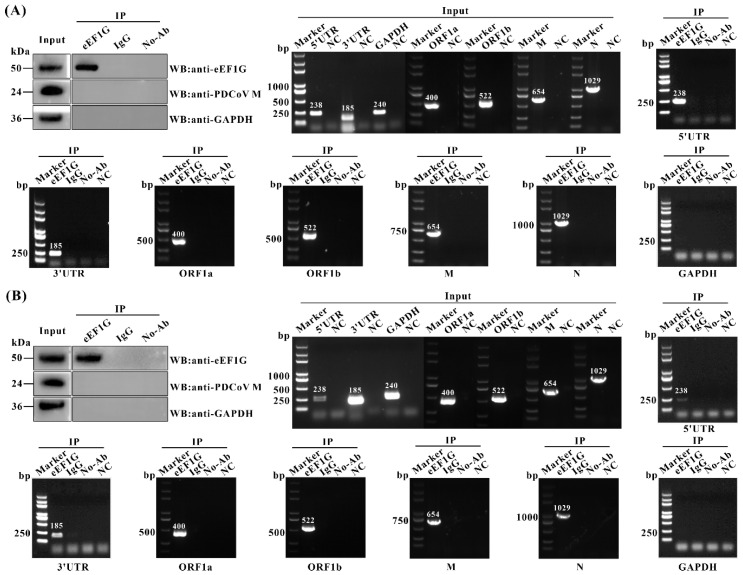
Cellular eEF1G binds to the PDCoV genomic RNA during infection. IPEC-J2 (**A**) and LLC-PK1 (**B**) cells were infected with PDCoV (MOI = 1) for 12 h and subsequently lysed using RIP lysis buffer. Following centrifugation, the clarified supernatant was collected and incubated with a rabbit anti-eEF1G pAb; parallel control samples were incubated with normal rabbit IgG or no antibody. The resulting antibody–antigen complexes were then immunoprecipitated with protein A/G magnetic beads. After five washes with RIP wash buffer, the beads were processed for either RNA extraction or Western blot analysis. Western blotting was performed using primary antibodies against the PDCoV M protein, eEF1G, and GAPDH (rabbit anti-eEF1G antibody was used). Co-precipitated RNA was extracted and analyzed by RT-PCR to detect specific regions of the PDCoV genome, including the 5′ UTR, 3′ UTR, ORF1a, ORF1b, M, and N genes, with GAPDH mRNA serving as a negative control. NC: PCR negative control.

**Figure 6 viruses-17-01369-f006:**
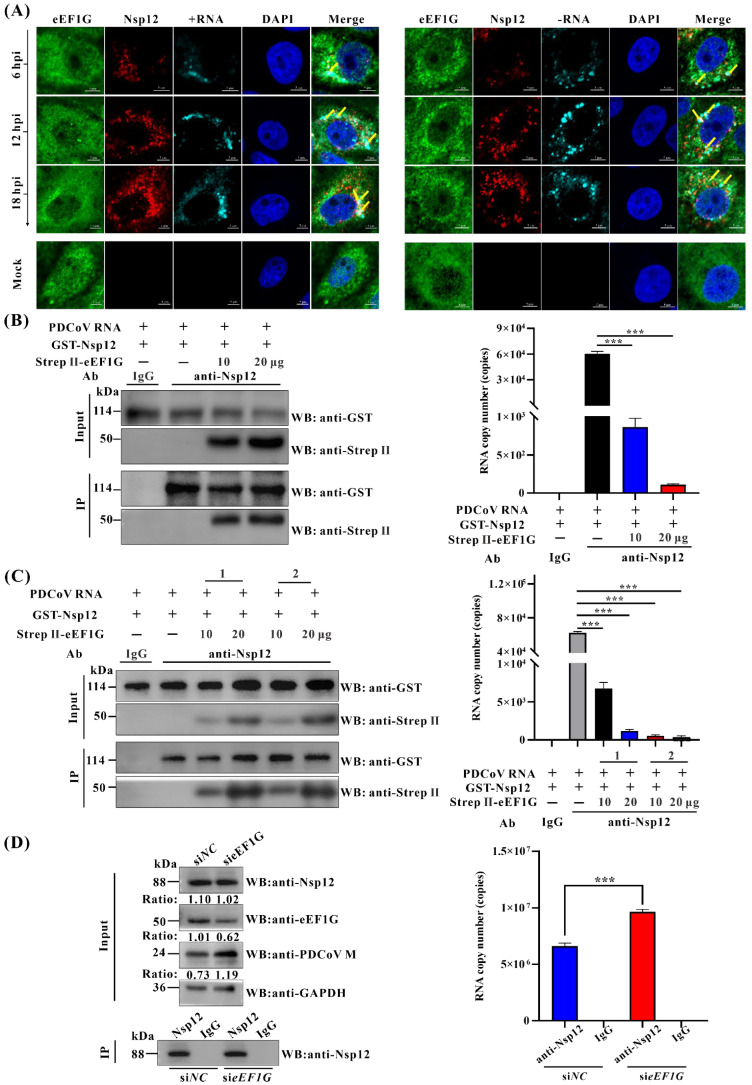
Cellular eEF1G interferes with the binding of Nsp12 to the PDCoV genomic RNA during infection. (**A**) IPEC-J2 cells were either mock-infected or infected with PDCoV (MOI = 1). At 6, 12, and 18 hpi, the cells were fixed and immunostained with primary antibodies against Nsp12 and eEF1G (mouse anti-eEF1G antibody was used), followed by corresponding Alexa Fluor 488- and 568-conjugated secondary antibodies. Subsequently, RNAscope was performed using probes targeting positive- (+) and negative-stranded (−) PDCoV RNA, which were then visualized using the TSA Vivid Fluorophore Kit 650. Using ImageJ software (version 2.16), the positive- and negative-stranded RNAs were pseudo-colored cyan. Nuclei were counterstained with DAPI. Pictures represent eEF1G (Green), PDCoV Nsp12 protein (Red), positive- (+) or negative-stranded (−) PDCoV RNA (Cyan), nuclei (Blue), and merged images (Merge). Colocalization is indicated by yellow arrows. Scale bar: 5 μm. (**B**) Ten micrograms of purified GST-Nsp12, 1 μg of total RNA extracted from PDCoV-infected cells, and either 10 μg or 20 μg of Strep II-eEF1G were combined in a binding reaction. The resulting protein–RNA complexes were immunoprecipitated using an anti-Nsp12 pAb and protein A/G magnetic beads, with normal rabbit IgG used as a negative control. Following five washes with RIP lysis buffer, the beads were collected for subsequent Western blot analysis and RNA extraction. The presence of GST-Nsp12 and Strep II-eEF1G in the precipitates was confirmed by Western blot. Co-precipitated RNA was extracted, and the amount of PDCoV RNA was quantified by absolute RT-qPCR targeting the PDCoV 5′ UTR. (**C**) Ten micrograms of purified GST-Nsp12 protein and 1 μg of total RNA extracted from PDCoV-infected IPEC-J2 cells were incubated with 10 μg or 20 μg of Strep II-eEF1G. Two experimental groups were established based on the order of addition: in Group 1, GST-Nsp12 was pre-incubated with Strep II-eEF1G before the addition of total RNA; in Group 2, total RNA was pre-incubated with Strep II-eEF1G prior to the addition of GST-Nsp12. The resulting protein–RNA complexes were immunoprecipitated using an anti-Nsp12 polyclonal antibody and protein A/G magnetic beads, with normal rabbit IgG serving as a negative control. After five washes with RIP lysis buffer, the beads were collected for subsequent Western blot analysis and RNA extraction. The presence of GST-Nsp12 and Strep II-eEF1G was confirmed by Western blot, and the associated RNA was quantified via absolute quantitative RT-qPCR targeting the PDCoV 5′ UTR. (**D**) IPEC-J2 cells were transfected with siRNAs for 36 h and subsequently infected with PDCoV at an MOI of 1 for 12 h. Cells were subsequently lysed and subjected to immunoprecipitation using an anti-Nsp12 antibody, with normal rabbit IgG serving as an isotype control. Immunoprecipitated complexes and whole-cell lysates were analyzed by Western blot to detect the presence of PDCoV M protein, Nsp12, eEF1G, and GAPDH (rabbit anti-eEF1G antibody was used). Viral RNA abundance was quantified by absolute quantitative RT-qPCR targeting the PDCoV 5′ UTR. Data are presented as the mean ± SD of three independent biological replicates and were analyzed using one-way ANOVA. *** *p* < 0.001.

**Figure 7 viruses-17-01369-f007:**
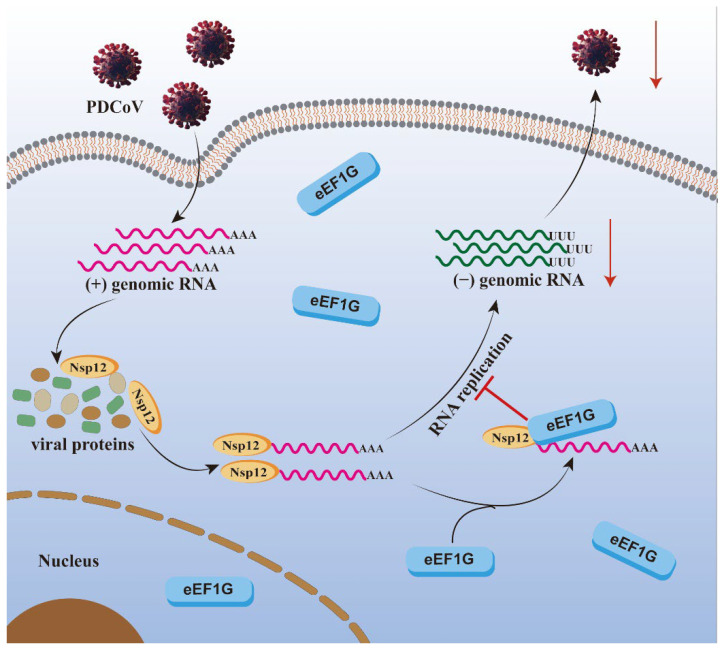
A proposed model illustrates the molecular mechanism by which cellular eEF1G negatively regulates PDCoV replication. Cellular eEF1G directly binds to the PDCoV RNA-dependent RNA polymerase, Nsp12. This interaction competitively hinders the binding of Nsp12 to the viral genomic RNA, thereby disrupting the synthesis of negative-stranded RNA and ultimately suppressing PDCoV replication. The red arrows represent decrease. Blocks in different colors represent different viral proteins.

## Data Availability

All the datasets supporting the conclusions of this article are included within the article or Appendix A.

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
