# Peer review of "Cellular eEF1G Inhibits Porcine Deltacoronavirus Replication by Binding Nsp12 and Disrupting Its Interaction with Viral Genomic RNA"

_viruses, 2025, doi:10.3390/v17101369_

Round 1
Reviewer 1 Report
Comments and Suggestions for Authors This study demonstrated host factor eEF1G interacts directly with PDCoV Nsp12 and binds viral genomic RNA, thereby disrupting the formation of the Nsp12–RNA complex, which provide novel insights into the role of host translational machinery in modulating coronavirus infection. While several questions should be cleared. 1. The lifecycle of PDCoV should be added to explain the sellected time points in the following experiments. 2. The expression level (including mRNA, protein, and distribution) of eEF1G during virus infection should detect. 3. eEF1G inhibit the synthesis of PDCoV positive-stranded RNA at 9 hpi and 12 hpi, although no significant differences. For host resist virus infection occurs in early infection, the roles of eEF1G in the early stage of PDCoV infection should explored. 4. Authors demonstrated eEF1G inhibiting viral infection by siRNA. Cellular eEF1G abundance and localization were unchanged during PDCoV infection at 12 hpi, 24 hpi, and 36 hpi. Whether exhibit other components or pathway that affect the function of eEF1G?This question can conclude by published document analysis. minor question Is porcine enteric alphacoronavirus (PEAV) right?Author Response
Responses to the reviewers’ comments and suggestions
Thank you very much for taking the time to review this manuscript. Please find the detailed responses below and the corresponding revisions in yellow marks in the re-submitted files.
Item 1: The lifecycle of PDCoV should be added to explain the sellected time points in the following experiments.
Response: We appreciate the reviewer's insightful suggestions. Our selection of time points was guided by both the established PDCoV replication cycle, which is completed in about 5-6 hours (Qin et al., 2019, Viruses 11:455), and our own characterization of the CHN-HN-1601 strain. Our one-step growth curve in LLC-PK1 cells shown in Figure S2 confirmed active replication, with viral titers rising from 6 hpi and peaking at 24 hpi, leading us to select the 6-24 hpi window for this cell line. However, we found that replication in IPEC-J2 cells is delayed. Since the viral M protein was not reliably detected by immunofluorescence until 12 hpi in these cells, we used the later window of 12-36 hpi for IPEC-J2 experiments to accurately capture the replication phase. In the new version of our manuscript, a description of the results was added to the section with a brief heading of “Cellular eEF1G Directly Interacts with the PDCoV Nsp12 Protein” on page 7, line 341. A reference (references 31 of the newly revised version) was added to meet the needs of revising our manuscript, the orders of partial references in our newly revised manuscript were rearranged accordingly.
Figure S2. One-step growth curve of PDCoV in LLC-PK1 cells. LLC-PK1 cells were infected with PDCoV at an MOI of 5, then harvested at 6, 12, 18, 24, 30, 36, 42, 48, 54, and 60 hpi. Viral yields were determined by TCID50 assay at different time points.
Reference:
Qin, P.; Du, E.Z.; Luo, W.T.; Yang, Y.L.; Zhang, Y.Q.; Wang, B.; Huang, Y.W. Characteristics of the life cycle of porcine deltacoronavirus (PDCoV) in vitro: Replication kinetics, cellular ultrastructure and virion morphology, and evidence of in-ducing autophagy. Viruses 2019, 11, 455. doi: 10.3390/v11050455.
Item 2: The expression level (including mRNA, protein, and distribution) of eEF1G during virus infection should detect.
Response: We thank the reviewer for this suggestion. Our analysis of eEF1G expression during PDCoV infection includes data at both the protein and mRNA levels. The protein expression and distribution, shown in Figure S6C, D, and F in the new version of the supplementary figures, are unaltered. Furthermore, RT-qPCR analysis (using β-actin for normalization) shown in Figure S6A and S6B confirms that eEF1G mRNA levels also remain stable throughout infection, consistent with the protein data. In the new version of our manuscript, a description of the results was added to the section with a brief heading of “eEF1G Binds to the Genomic RNA of PDCoV during Infection” on page 12, line 489. The Discussion section was added on page 18, line 682.
Figure S6. Cellular eEF1G abundance and localization are unchanged by PDCoV infection or Nsp12 expression. (A) IPEC-J2 cells were either mock-infected or infected with PDCoV (MOI=1), then harvested at 12, 24, and 36 hpi. The eEF1G mRNA levels were quantified by RT-qPCR using β-actin as a reference gene. Relative RNA levels at each time point were normalized to the mock-infected control. Data, presented as means ± SD from three independent experiments, were analyzed by two-way ANOVA. ns, no significance. (B) LLC-PK1 cells were infected and analyzed as described in (A).
Item 3: eEF1G inhibit the synthesis of PDCoV positive-stranded RNA at 9 hpi and 12 hpi, although no significant differences. For host resist virus infection occurs in early infection, the roles of eEF1G in the early stage of PDCoV infection should explored.
Response: We thank the reviewer for their helpful suggestion. As coronaviruses enter cells via receptor-mediated endocytosis and begin replication upon the release of genomic RNA into the cytoplasm, we re-evaluated the potential role of eEF1G in these early stages of PDCoV infection. Our results confirm that eEF1G does not influence the initial phases of the viral lifecycle. Specifically, RT-qPCR analyses showed no significant effect of eEF1G knockdown on viral adsorption or internalization (Figure 4A, B). Furthermore, eEF1G knockdown did not alter the expression level of the Nsp12 protein (Figure 6D), indicating it does not affect the translation of viral proteins. Therefore, we conclude that the inhibitory effect of eEF1G occurs after viral entry and initial protein synthesis. Instead, as we have demonstrated, eEF1G specifically functions to inhibit the synthesis of PDCoV negative-stranded RNA. In response to the reviewer's comment, we have added detailed descriptions of the adsorption/internalization protocols and these results to the respective sections of the manuscript. The Discussion section was added on page 18, line 664–672.
Item 4: Authors demonstrated eEF1G inhibiting viral infection by siRNA. Cellular eEF1G abundance and localization were unchanged during PDCoV infection at 12 hpi, 24 hpi, and 36 hpi. Whether exhibit other components or pathway that affect the function of eEF1G?This question can conclude by published document analysis.
Response: We thank the reviewer for their question. eEF1G is a multifunctional protein involved in translation, cellular stress responses, and redox reactions. However, the upstream pathways that regulate eEF1G's function remain poorly characterized. Structurally, eEF1G is a core component of the eukaryotic elongation factor 1 (eEF1) complex, where it serves as the γ subunit of the eEF1B guanine nucleotide exchange subcomplex. Within this subcomplex—comprising eEF1Bα, eEF1Bβ, and eEF1G—eEF1G is thought to act as a structural scaffold essential for its assembly. A study by Achilonu et al. (2018) demonstrated that eEF1Bβ interacts with eEF1G with high functional affinity, suggesting that eEF1Bβ may modulate the formation of an ordered eEF1G oligomer, a structure potentially critical for protein biosynthesis. Beyond its role in the complex, eEF1G function can also be modulated by post-translational modification. For instance, its C-terminus contains a conserved LAMMER kinase phosphorylation site that is important for its normal function (Fan et al., 2010). The Discussion section was added on pages 17–18, line 638–645. Two references (references 39, 40 of the newly revised version) were added to meet the needs of revising our manuscript, the orders of partial references in our newly revised manuscript were rearranged accordingly.
Reference:
Achilonu I.; Elebo N.; Hlabano B.; Owen G.R.; Papathanasopoulos M.; Dirr H.W. An update on the biophysical character of the human eukaryotic elongation factor 1 beta: Perspectives from interaction with elongation factor 1 gamma. J Mol Recognit 2018, 31, e2708. doi: 10.1002/jmr.2708.
Fan Y.; Schlierf M.; Gaspar A.C.; Dreux C.; Kpebe A.; Chaney L.; Mathieu A.; Hitte C.; Grémy O.; Sarot E.; Horn M.; Zhao Y.; Kinzy T.G.; Rabinow L. Drosophila translational elongation factor-1gamma is modified in response to DOA kinase activity and is essential for cellular viability. Genetics 2010, 184, 141–154. doi: 10.1534/genetics.109.109553.
Item 5: Is porcine enteric alphacoronavirus (PEAV) right?
Response: The porcine enteric alphacoronavirus (PEAV), also referred to as swine acute diarrhea syndrome coronavirus (SADS-CoV), was first identified in 2017. However, a standardized nomenclature for this pathogen is still lacking. To ensure clarity and align with the terminology used in a growing body of contemporary research (Chen et al., 2023; Wang et al., 2023; Xu et al., 2020), we have selected "porcine enteric alphacoronavirus" (PEAV) for this manuscript.
Reference:
Chen X.N.; Liang Y.F.; Weng Z.J.; Quan W.P.; Hu C.; Peng Y.Z.; Sun Y.S.; Gao Q.; Huang Z.; Zhang G.H.; Gong L. Porcine enteric alphacoronavirus entry through multiple pathways (caveolae, clathrin, and macropinocytosis) requires Rab GTPases for endosomal transport. J Virol 2023, 97, e0021023. doi: 10.1128/jvi.00210-23.
Wang W.; Zhou L.; Ge X.; Han J.; Guo X.; Zhang Y.; Yang H. Analysis of codon usage patterns of porcine enteric alphacoronavirus and its host adaptability. Virology 2023, 587, 109879. doi: 10.1016/j.virol.2023.109879.
Xu Z.; Gong L.; Peng P.; Liu Y.; Xue C.; Cao Y. Porcine enteric alphacoronavirus inhibits IFN-α, IFN-β, OAS, Mx1, and PKR mRNA expression in infected Peyer's patches in vivo. Front Vet Sci 2020, 7, 449. doi: 10.3389/fvets.2020.00449.

Reviewer 2 Report
Comments and Suggestions for Authors
Suggestions
September 23, 2025
The paper “Cellular eEF1G inhibits porcine deltacoronavirus replication by binding Nsp12 and disrupting its interaction with viral genomic RNA” submitted to Viruses has demonstrated that eEF1G, a central component of the host translational machinery, acts as a novel host restriction factor for PDCoV. Mechanistically, eEF1G interacts directly with PDCoV Nsp12 and inhibits PDCoV replication by competing with Nsp12 for genomic RNA binding, ultimately inhibiting negative-stranded RNA synthesis. The findings of this study provide new insights into the role of eEF1G in viral replication and elucidate the molecular mechanisms. The manuscript is well-written that only needs to undergo a few minor changes. Some questions and issues should be carefully addressed before recommending this manuscript for publication. There are some specific suggestions:
- Has the author verified that the eEF1G does not affect viral adsorption and internalization?
- It is best to label the proteins represented by each fluorescent channel in the top row of the images.
- In Figure 2I, the author should label the names of the two proteins in the picture.
- Line 166: “–ΔΔCT” should be superscripted.
- Due to the homologous of the viral proteins and genomes in Coronaviridae, have the authors conducted researches on other coronaviruses?
- The uniqueness of this study compared to other researches which demonstrate eEF1G promotes viral replication requires more discussion.
- In Figure S4D, why did the author stain dsRNA?
Author Response
Responses to the reviewers’ comments and suggestions
Thank you very much for taking the time to review this manuscript. Please find the detailed responses below and the corresponding revisions in yellow marks in the re-submitted files.
Item 1: Has the author verified that the eEF1G does not affect viral adsorption and internalization?
Response: Thank you for this insightful question. In order to address this question, we have reanalyzed the effect of the eEF1G on both viral adsorption and internalization of PDCoV, using RT-qPCR analyses. The corresponding results are shown in Figure 4A and B. No significant differences in the levels of viral adsorption or internalization were observed between the sieEF1G and siNC groups, indicating that the eEF1G does not affect viral adsorption and internalization. Furthermore, a description of the protocols was added to the section with a brief heading of “Viral Adsorption and Internalization Assay”on page 6, line 280. A description of the results was added to the section with a brief heading of“eEF1G Functions to Inhibit the Synthesis of PDCoV Negative-Stranded RNA”on page 11, line 456.
Item 2: It is best to label the proteins represented by each fluorescent channel in the top row of the images.
Response: We appreciate your suggestion. In the revised manuscript, the indicated fluorescent channels of Figure 2B, 6A, S1, S6F, and S7 were labeled in the top row of the images.
Item 3: In Figure 2I, the author should label the names of the two proteins in the picture.
Response: We appreciate your suggestion. In the revised manuscript, we labeled the eEF1G and Nsp12 in Figure 2I.
Item 4: Line 166: “–ΔΔCT” should be superscripted.
Response: We appreciate your careful review. In the revised manuscript, we superscripted the “–ΔΔCT”.
Item 5: Due to the homologous of the viral proteins and genomes in Coronaviridae, have the authors conducted researches on other coronaviruses?
Response: We thank the reviewer for this helpful question. In response, we have now investigated the role of eEF1G in the replication of two additional coronaviruses: PEDV (BJ2011C, GenBank: KX066126.1) and PEAV (GDZQ-2018, GenBank: MW727454.1). Immunofluorescence assays revealed that the replication of both viruses was significantly enhanced in eEF1G-knockdown cells compared to the siNC control (Figure S4A, C). This was evidenced by a marked increase in the number of cells positive for the PEAV N and PEDV N proteins, respectively (Figure S4B, D). These consistent findings across PDCoV, PEDV, and PEAV suggest that eEF1G may inhibit viral replication through a conserved mechanism common to these coronaviruses. In the new version of our manuscript, a description of the materials was added to the section with a brief heading of “Cells and Virus” on page 3, line 115. A description of the results was added to the section with a brief heading of “Cellular eEF1G Acts as a Host Restriction Factor for PDCoV” on page 10, line 419. The Discussion section was added on page 18, line 653.
Figure S4. Knockdown of eEF1G significantly promotes PEAV and PEDV replication. (A) ST cells were transfected with either sieEF1G or scrambled control siRNA (siNC) for 36 h and then infected with PEAV at an MOI of 1. At 12 hpi, the cells were fixed and processed for confocal immunofluorescence analysis. After incubation with a primary antibody against PEAV N protein, the cells were treated with an Alexa Fluor 488-conjugated secondary antibodies. Nuclei were counterstained with DAPI. Scale bar: 200 μm. (B) Statistical analysis of the percentage of N protein-positive cells among PEAV-infected ST cells shown in (A). *** p < 0.001. (C) MARC-145 cells were transfected with either sieEF1G or scrambled control siRNA (siNC) for 36 h and then infected with PEDV at an MOI of 1. At 12 hpi, the cells were fixed and processed for confocal immunofluorescence analysis. After incubation with a primary antibody against PEDV N protein, the cells were treated with an Alexa Fluor 488-conjugated secondary antibodies. Nuclei were counterstained with DAPI. Scale bar: 200 μm. (D) Statistical analysis of the percentage of N protein-positive cells among PEDV-infected MARC-145 cells shown in (C). *** p < 0.001.
Item 6: The uniqueness of this study compared to other researches which demonstrate eEF1G promotes viral replication requires more discussion.
Response: We thank the reviewer for this useful suggestion. Indeed, as noted, several studies have implicated eEF1G in various stages of viral replication: Warren et al. reported its interaction with the HIV-1 reverse transcriptase p51 subunit to promote reverse transcription. Sammaibashi et al. showed its role in IAV protein translation and as a potential scaffold for the viral polymerase. Gallie et al. demonstrated its binding to TBSV RNA to facilitate minus-strand synthesis as an RNA chaperone. Our study builds upon this foundation but provides several key advances in the context of coronavirus infection. Using a multifaceted approach (western blot, immunofluorescence, viral titer assays), we not only confirm an interaction between eEF1G and the PDCoV polymerase Nsp12 and demonstrate its RNA-binding capacity—findings consistent with the broader role of eEF1G. However, our work reveals a novel, inhibitory function. We found that eEF1G inhibits negative-strand RNA synthesis by obstructing the initial binding of the RdRp to the genomic template. This contrasts with the proviral roles described in other systems and highlights that eEF1G can regulate viral RNA synthesis through distinct, virus-specific mechanisms at different phases of the replication cycle. The Discussion section was rewritten on page 18, line 647–650 and 674–677.
Reference:
Warren, K.; Wei, T.; Li, D.; Qin, F.; Warrilow, D.; Lin, M.-H.; Sivakumaran, H.; Apolloni, A.; Abbott, C.M.; Jones, A.; et al. Eukaryotic elongation factor 1 complex subunits are critical HIV-1 reverse transcription cofactors. Proc Natl Acad Sci U S A 2012, 109, 9587–9592, doi: 10.1073/pnas.1204673109.
Sammaibashi, S.; Yamayoshi, S.; Kawaoka, Y. Strain-specific contribution of eukaryotic elongation factor 1 gamma to the translation of influenza A virus proteins. Front Microbiol 2018, 9, 1446, doi: 10.3389/fmicb.2018.01446.
Gallie, D.; Sasvari, Z.; Izotova, L.; Kinzy, T.G.; Nagy, P.D. Synergistic roles of eukaryotic translation elongation factors 1Bγ and 1A in stimulation of tombusvirus minus-strand synthesis. PLoS Pathog 2011, 7, e1002438, doi: 10.1371/journal.ppat.1002438.
Item 7: In Figure S4D, why did the author stain dsRNA?
Response: We thank the reviewer for this valuable question. As a well-established marker for the sites of viral RNA synthesis, double-stranded RNA (dsRNA) has been widely used to identify viral replication complexes. This approach was pioneered in studies of SARS-CoV, where dsRNA was found to localize primarily to the interior of double-membrane vesicles (DMVs) (Knoops et al., 2008, PLoS Biol 6: e226). The method has since been adopted in numerous other studies on coronavirus replication (e.g., Hagemeijer et al., 2012; Deng et al., 2017; Klein et al., 2020). Therefore, to investigate whether eEF1G translocates to PDCoV replication sites, we used dsRNA staining as a reliable marker.
Reference:
Knoops K.; Kikkert M.; Worm S.H.; Zevenhoven-Dobbe J.C.; van der Meer Y.; Koster A.J.; Mommaas A.M.; Snijder E.J. SARS-coronavirus replication is supported by a reticulovesicular network of modified endoplasmic reticulum. PLoS Biol 2008, 6, e226, doi: 10.1371/journal.pbio.0060226.
Hagemeijer M.C.; Rottier P.J.; de Haan C.A. Biogenesis and dynamics of the coronavirus replicative structures. Viruses 2012, 4, 3245–3269. doi: 10.3390/v4113245.
Deng X.; Hackbart M.; Mettelman R.C.; O'Brien A.; Mielech A.M.; Yi G.; Kao C.C.; Baker S.C. Coronavirus nonstructural protein 15 mediates evasion of dsRNA sensors and limits apoptosis in macrophages. Proc Natl Acad Sci U S A 2017, 114, E4251–E4260. doi: 10.1073/pnas.1618310114.
Klein S.; Cortese M.; Winter S.L.; Wachsmuth-Melm M.; Neufeldt C.J.; Cerikan B.; Stanifer M.L.; Boulant S.; Bartenschlager R.; Chlanda P. SARS-CoV-2 structure and replication characterized by in situ cryo-electron tomography. Nat Commun 2020, 11, 5885. doi: 10.1038/s41467-020-19619-7.
